# Ultrasound-Assisted Mineralization of 2,4-Dinitrotoluene in Industrial Wastewater Using Persulfate Coupled with Semiconductors

**DOI:** 10.3390/molecules28114351

**Published:** 2023-05-25

**Authors:** Wen-Shing Chen, Min-Chih Hsu

**Affiliations:** Department of Chemical and Materials Engineering, National Yunlin University of Science & Technology, Yunlin, Douliou 64002, Taiwan

**Keywords:** dinitrotoluene, persulfate, semiconductor, sulfate radical, ultrasound

## Abstract

Oxidative degradation of 2,4-dinitrotoluenes in aqueous solution was executed using persulfate combined with semiconductors motivated by ultrasound (probe type, 20 kHz). Batch-mode experiments were performed to elucidate the effects of diverse operation variables on the sono-catalytic performance, including the ultrasonic power intensity, dosage of persulfate anions, and semiconductors. Owing to pronounced scavenging behaviors caused by benzene, ethanol, and methanol, the chief oxidants were presumed to be sulfate radicals which originated from persulfate anions, motivated via either the ultrasound or sono-catalysis of semiconductors. With regard to semiconductors, the increment of 2,4-dinitrotoluene removal efficiency was inversely proportional to the band gap energy of semiconductors. Based on the outcomes indicated in a gas chromatograph–mass spectrometer, it was sensibly postulated that the preliminary step for 2,4-dinitrotoluene removal was denitrated into *o*-mononitrotoluene or *p*-mononitrotoluene, followed by decarboxylation to nitrobenzene. Subsequently, nitrobenzene was decomposed to hydroxycyclohexadienyl radicals and converted into 2-nitrophenol, 3-nitrophenol, and 4-nitrophenol individually. Nitrophenol compounds with the cleavage of nitro groups synthesized phenol, which was sequentially transformed into hydroquinone and *p*-benzoquinone.

## 1. Introduction

2,4-dinitrotoluene is a vital raw material used primarily for the manufacture of tolylene diisocyanate, an important petrochemical intermediate in the synthetic leather industry. Its derivatives are extensively applied in the preparation of rubbers, paints, and dyes [1]. However, it has been specified to be one of the prior contaminants to be treated by the US Environmental Protection Agency [2]. Factory effluents polluted with 2,4-dinitrotoluene should be disposed of on account of their serious toxicity and carcinogenic properties [3]. There is an increasing trend of the generation of related wastes from the activities of these industries [4]. Consequently, the mineralization of 2,4-dinitrotoluene has attracted much attention in the last decade. 

Up to now, several publications have focused on the sonochemical decomposition of 2,4-dinitrotoluene [5,6,7]. Thermal pyrolysis is proposed to be the main degradation mechanism of 2,4-dinitrotoluene oxidation, wherein sonolytic temperature is the most significant influencing factor. The sonochemical destruction of 2,4-dinitrotoluene could be obviously accelerated with the assistance of TiO_2_ powder, which supplied extra nuclei for the generation of cavitation microbubbles. In our previous work, it was found that the removal rate of 2,4-dinitrotoluene using a traditional electro-Fenton method could increase significantly through the use of ultrasonic irradiation due to the enhancement of the mass transfer rate of oxygen toward the cathode for the production of hydrogen peroxide [8]. The other conventional methods for the treatment of nitroaromatics include TiO_2_/UV [9,10], Fenton’s reagent/UV [11,12], O_3_ [13], H_2_O_2_/O_3_, and UV/O_3_ methods [14], wherein hydroxyl radicals and holes were considered to be principal oxidants [15,16]. Nonetheless, increasing interest has been paid to sulfate-radical-based processes because of their higher redox potential (2.6–3.1 V) and the long lifetime of sulfate radicals in wastewater [17,18]. Sulfate radicals can be generated via persulfate anions motivated by various manners, such as thermal energy, ultraviolet irradiation (Equation (1)), electron donation of transition metal ions (Equation (2)) [19,20,21], electrochemical reduction by cathode (Equation (3)) [22], and ultrasonic irradiation (Equation (4)) [23].
S_2_O_8_^2−^ + Thermal Energy/UV → 2 SO_4_^•−^(1)
S_2_O_8_^2−^ + Metal^n+^ → SO_4_^•−^ + SO_4_^2−^ + Metal^(n+1)+^(2)
S_2_O_8_^2−^ + e^−^ → SO_4_^•−^ + SO_4_^2−^(3)
S_2_O_8_^2−^ + US → 2SO_4_^•−^(4)

US denotes ultrasonic irradiation.

It has been recognized that the sono-activated persulfate process is effective for the oxidative degradation of 2,4-dinitrotoluene, wherein some persulfate anions are transformed into reactive sulfate radicals via cavitation microbubbles [24,25]. The removal efficiency of 2,4-dinitrotoluene could increase upon the addition of electrolytes, which would inhibit the coalescence of tiny cavitation microbubbles, leading to a higher cavitation strength [26]. Persulfate combined with zero-valent iron (Fe^0^) was used to dispose 2,4-dinitrotoluene, of which a reduction in nitro groups and a sequential sulfate radical oxidation mechanism was proposed [27]. An analogous announcement on the reduction in the nitro group of 2,4-dinitrotoluene to an amino group, resulting in an enhancement of its removal rate, was also reported for the process of persulfate coupled with iron sulfide [28]. Additionally, the decomposition of 2,4-dinitrotoluene was performed using persulfate activated by Fe^2+^ released from the electrolysis of iron metal integrated with active carbon [29,30]. Fe-Mn binary oxide was also verified for the effective transformation of persulfate into sulfate radicals via electrons transfer over active sites of Fe^2+^, Mn^2+^, and Mn^3+^ [31]. Nonetheless, natural aquatic constituents, including HCO_3_^−^, CO_3_^2−^, and NO_3_^−^, exhibited negative effects on the degradation of 2,4-dinitrotoluene in the persulfate/Fe^2+^ process [32].

Recently, a promising process has been developed on persulfate motivated simultaneously with ultrasound and heterogeneous semiconductors, which were stimulated by sonoluminescence emitted from cavitation microbubbles [33,34]. The light energy of sonoluminescence meets the requirement for the band gap energy of the semiconductors, leading to the generation of electron–hole pairs. The sono-induced electrons of the semiconductors would convert persulfate anions into sulfate radicals. In fact, it has been applied successfully for the mineralization of aniline [35] and nitrobenzene [36], whereas the systematic investigation of semiconductor species commonly applied is scarce. Consequently, this research is devoted to making clear the relationships between heterogeneous semiconductors and ultrasonic irradiation via the mineralization of 2,4-dinitrotoluene in aqueous solution using persulfate integrated with a variety of semiconductors. A series of semiconductors were examined, including ZnO [37], Ni_2_O_3_ [38], NiO [39], SnO_2_ [40], Fe_2_O_3_ [41], Cu_2_O [42], TiO_2_ [43], PbO [44], and Pb_3_O_4_ [45]. Experiments on the synthetic Ag (1–4 wt%)/NiO semiconductors with distinct band gap energy were also conducted, wherein the influential operating variables on the degradation behaviors of 2,4-dinitrotoluene were investigated, such as the ultrasonic power intensity, persulfate concentrations, and dosage of Ag/NiO. The 2,4-dinitrotoluene degradation pathways imposed by persulfate combined with semiconductors under ultrasonic irradiation were cautiously determined as well.

## 2. Results and Discussion

### 2.1. Comparison of Persulfate Oxidation Simply and Persulfate Integrated with Diverse Semiconductors under Ultrasonic Irradiation

Figure 1a illustrates the time-dependent drawings of the TOC removal efficiency executed by persulfate oxidation and persulfate integrated with various semiconductors under ultrasonic irradiation, respectively. Transparently, 2,4-dinitrotoluene removal rates using persulfate integrated with semiconductors motivated by ultrasound were much faster than those achieved by utilizing persulfate oxidation alone. This observation may be interpreted with an increase in the yield of highly reactive sulfate radicals (E° = 2.6 V), wherein °°persulfate anions (E° = 2.01 V) could be motivated either by ultrasound [21] or via the sono-catalysis of semiconductors [46,47]. Particularly, it is likely that the increment in the TOC removal efficiency between the presence and absence of semiconductors could be associated with the band gap energy of semiconductors, including SnO_2_ (3.9 eV) [38], Ni_2_O_3_ (3.67 eV) [48], ZnO (3.37 eV) [49], NiO (3.23 eV) [50], TiO_2_ (P-25) (3.1 eV) [51], Pb_3_O_4_ (2.2 eV) [52], Fe_2_O_3_ (2.2 eV) [53], Cu_2_O (2.15 eV) [54], and PbO (1.99 eV) [55]. Low band gap energy is beneficial for the oxidative removal of 2,4-dinitrotoluene (refer to Figure 1b). That is, the sono-catalytic performance is significantly promoted over semiconductors with low band gap energy, wherein the reactions which occurred are listed as follows.
Semiconductor + US → h^+^_vb_ + e^−^_cb_(5)
S_2_O_8_^2−^ + e^−^_cb_→ SO_4_^•−^ + SO_4_^2−^(6)
SO_4_^2−^ + h^+^_vb_→ SO_4_^•−^(7)

The symbol of h^+^_vb_ stands for sono-induced holes in the valence band and e^−^_cb_ stands for sono-induced electrons in the conduction band.

### 2.2. Effect of Band Gap Energy on Persulfate Integrated with Ag/NiO Semiconductor

To disclose the relationship between the sono-catalytic activity and band gap energy of semiconductors, Ag (1–4 wt%)/NiO semiconductors were prepared, examined, and tested. Appendix A demonstrates the FE-SEM images of Ag (1–4 wt%)/NiO semiconductors. It indicates that most of the surface of NiO was smooth, coated with some irregularly shaped particulates. As the loading amounts of Ag increased, more clumps of particulates were observed. Table 1 presents EDS element analyses of Ag (1–4 wt%)/NiO semiconductors. The real weight percentage of Ag measured was in agreement with that impregnated theoretically. Figure 2 illustrates X-ray diffraction patterns of Ag (1–4 wt%)/NiO semiconductors. The four particular diffraction peaks at 2θ values of 37.5°, 43.8°, 63.9°, and 77.4°, corresponding to the planes of (111), (200), (220), and (311), respectively, possess characteristic peaks of the face-centered cubic structure of metallic silver [56,57]. This provides more evidence for the existence of Ag metal loaded on the surface of NiO.

Figure 3 depicts the UV-vis diffuse reflectance spectra of Ag/NiO semiconductors. The NiO spectrum exhibits two main peaks at wavelengths of 370 and 390 nm, corresponding to band gap energy of 3.23 eV [50]. Instead, the spectra of Ag/NiO present two sharp peaks at wavelengths of 380 and 400 nm, followed with a broad peak between the wavelength of 460 and 610 nm. This implies that Ag/NiO responds strongly to visible light. This phenomenon could be attributed to the loading of Ag metal, which furnished an electron sink and prevented the recombination of sono-induced electrons with holes, leading to an enhancement of the yield of conductive electrons descended from NiO [58,59,60]. Likewise, the band gap energy of Ag/NiO was evaluated on the basis of Tauc’s relation ((αhν)^1/n^ = A(hν − E_g_)), in which hν manifests incident photo energy. The “n” value was usually set at a value of 2 or 1/2, depending on whether its electronic transition was in either a direct or indirect state. The variation in (αhν)^2^ was portrayed versus photo energy (hν) in a graph, wherein the band gap energy was estimated from the intercept of the tangent line to the X-axis (see Appendix A) [61,62,63,64]. Accordingly, the band gap energy of Ag (1–4 wt%)/NiO was determined between 3.15 and 2.53 eV (refer to Table 1), consistent with the results of Pandey et al. [65]. As expected, Ag (1–4 wt%)/NiO semiconductors with lower band gap energy were favorable for the removal of 2,4-dinitrotoluene compared with NiO. The increment of the TOC removal efficiency in comparison to the presence or absence of Ag (1–4 wt%)/NiO was inversely proportional to the band gap energy (see Figure 4), consistent with the results demonstrated in Figure 1b. The superior sono-catalytic performance of semiconductors with lower band gap energy was excited more easily by sonoluminescence and could be ascribed to the increased enhancement of the electric charge conduction. It is likely that sonoluminescence could provoke them to generate electron–hole pairs. The sono-induced electrons may transform persulfate anions into sulfate radicals; likewise, sono-induced holes may convert sulfate anions into sulfate radicals (see Equations (6) and (7)). Because of the higher removal efficiency of 2,4-dinitrotoluene, Ag (4 wt%)/NiO was elected as a candidate for further testing. 

XPS analyses were executed to examine the intrinsic electronic state of Ag (4 wt%)/NiO. Figure 5 illustrates the Ni 2p XPS spectra of original Ag (4 wt%)/NiO and Ag (4 wt%)/NiO reacted. As far as original Ag (4 wt%)/NiO is concerned, four peaks centered at 851, 859, 870, and 877 eV were observed, which were separately assigned to the binding energy of Ni 2p_(3/2)_ and Ni 2p_(1/2)_ [66,67]. Nonetheless, the binding energy of Ni 2p_(3/2)_ and Ni 2p_(1/2)_ of Ag (4 wt%)/NiO shifted to 852 and 860 eV and 871 and 878 eV, respectively, after sono-catalytic experiments. It is evident that some Ni^2+^ states on the surface of Ag (4 wt%)/NiO have shifted to Ni^3+^ states relative to the original one [38] due to the migration of the sono-induced electrons generated. The outcomes verify a previous hypothesis that persulfate anions could be motivated into sulfate radicals via sono-induced electrons descended from semiconductors irradiated by ultrasound. Additionally, Ag (4 wt%)/NiO may convert fractional sulfate anions into sulfate radicals via sono-induced holes, which make a minor contribution towards the oxidation of 2,4-dinitrotoluene.

### 2.3. Effect of Scavenger Dosages on Persulfate Integrated with Ag (4 wt%)/NiO under Ultrasonic Irradiation

The identical concentration of scavengers, including benzene, ethanol, and methanol, was added to the aqueous solution, respectively, to evaluate the main reactive radicals for 2,4-dinitrotoluene oxidation in the process of persulfate coupled with Ag (4 wt%)/NiO motivated by ultrasound. As presented in Figure 6, the degradation rate of 2,4-dinitrotoluene was severely suppressed due to the existence of benzene, which reacted to sulfate radicals quickly at the high rate constant of 3 × 10^9^ M^−1^ s^−1^ [17]. On the other hand, the 2,4-dinitrotoluene decomposition rate was moderately inhibited by the presence of ethanol or methanol, which reacts to sulfate radicals under rate constants of 7.7 × 10^7^ M^−1^ s^−1^ and 3.2 × 10^6^ M^−1^ s^−1^ independently [68]. The descent of the 2,4-dinitrotoluene removal efficiency was consistent with the scavenging ability of sulfate radicals. The outcomes expose that sulfate radicals, which originated from persulfate anions, were dominant oxidants at the conditions of persulfate combined with Ag (4 wt%)/NiO irradiated by ultrasound.

### 2.4. Effect of Ultrasonic Power Intensity on Persulfate Integrated with Ag (4 wt%)/NiO under Ultrasonic Irradiation

It has been recognized that the optimization of ultrasonic power is an important issue for process design. Figure 7a illustrates the time-dependent drawings of the TOC removal efficiency under a variety of ultrasonic power intensities. Obviously, the 2,4-dinitrotoluene removal rate increased upon an increase in the ultrasonic power intensity. The yield of sulfate radicals would be expected to significantly increase, caused by the intense motivation of persulfate through ultrasonic irradiation or the sono-catalysis of Ag (4 wt%)/NiO semiconductors. Contrarily, the 2,4-dinitrotoluene decomposition rate was restrained at the highest power intensity (260 W cm^−2^). The phenomenon at extreme conditions could be attributed to a severe rise in the power intensity, leading to the acute generation of cavitation microbubbles, which would probably coalesce into vast bubbles and give rise to a decline in the cavitation strength. Accordingly, it would result in the formation of lower sulfate radical yields, corresponding to a lesser extent of benzene scavenging effects (shown in Figure 7b). The experimental outcomes coincide with the announcement of Sivakumar et al. [69,70], wherein the optimization of the ultrasonic power intensity has been found to be useful for the degradation of rhodamine and nitrophenol. In addition, the amount of oxygen gas dissolved in wastewater would significantly decrease due to the degassing effects caused by ultrasound [8]. Thus, the quantity of hydroxyl radicals descended from oxygen gas and sono-induced electrons on Ag (4 wt%)/NiO would transparently diminish (shown in Equation (8)). As a rule, it convinces us that sulfate radicals play dominant roles for 2,4-dinitrotoluene elimination. O_2_ + 2H_2_O + 3e^−^_cb_ → 3OH^−^ + ^•^OH(8)

### 2.5. Effect of Persulfate Concentrations on Persulfate Integrated with Ag (4 wt%)/NiO under Ultrasonic Irradiation

Based on economic viewpoints, it is essential to establish an optimal persulfate concentration on persulfate integrated with Ag (4 wt%)/NiO motivated by ultrasound. Figure 8a presents the time-dependent drawings of the TOC removal efficiency at different persulfate concentrations. Apparently, the 2,4-dinitrotoluene degradation rate displays an increasing trend upon raising persulfate concentrations. The events could be reasonably ascribed to a high sulfate radical yield descended from high concentrations of persulfate anions. Conversely, the mineralization rate of 2,4-dinitrotoluene decreased as there was an excess persulfate concentration (70 mM) by virtue of unexpected side reactions among persulfate residuals and sulfate radicals [71,72] due to a higher persulfate concentration being detected at the end of testing [73]. For the sake of elucidating the correlation between sulfate radical yields and benzene scavenging effects, experiments associated with or without benzene were performed independently (refer to Figure 8b). This clearly indicates that benzene scavenging effects exhibit an identical inclination with both sulfate radical yields and the TOC removal efficiency. Especially, the results outlined above confirm the inference that sulfate radicals were the principal oxidants. 

### 2.6. Effect of Dosages of Ag (4 wt%)/NiO Integrated with Persulfate under Ultrasonic Irradiation

A crucial dosage of Ag (4 wt%)/NiO for the promotion of 2,4-dinitrotoluene removal rates is necessary. Figure 9a demonstrates the influence of the addition of Ag (4 wt%)/NiO on the TOC removal efficiency. It shows distinctly that 2,4-dinitrotoluene decomposition rates rise with an increase in Ag (4 wt%)/NiO dosages, whereas an excess dosage of Ag (4 wt%)/NiO (≥1.5 g L^−1^) causes an unexpected conflict. The enhancement of 2,4-dinitrotoluene degradation rates could be ascribed to higher sulfate radical yields which originated from persulfate anions motivated by sono-induced electrons over Ag (4 wt%)/NiO (refer to Equation (6)). Nonetheless, the surplus amount of Ag (4 wt%)/NiO powder (≥1.5 g L^−1^) could attenuate the ultrasonic wave propagation, bringing about feebleness in the cavitation strength [7,74]. An identical trend was also observed on both the TOC removal efficiency and benzene scavenging effect (see Figure 9b). The results offer additional evidence that sulfate radicals are mainly responsible for the oxidation of 2,4-dinitrotoluene. It is notable that 2,4-dinitrotoluene contaminants can be totally eliminated under conditions of ultrasonic power intensity = 220 W cm^−2^, T = 318 K, persulfate concentration = 60 mM, and Ag (4 wt%)/NiO dosage = 1.35 g L^−1^. Particularly, under silent conditions, the 2,4-dinitrotoluene removal efficiency reached about 26%, which is lower than that achieved by the utilization of sono-activated persulfate oxidation alone (Figure 1a). This supports that persulfate anions could be excited either by ultrasound or via sono-induced electrons on semiconductors into sulfate radicals. In the research, the sono-catalytic stability of Ag(4 wt%)/NiO was examined via repetitions of six tests (shown in Figure 10). Evidently, the 2,4-dinitrotoluene removal efficiency reached nearly 98% during the whole experiment. This convinces us of the promising application of Ag/NiO semiconductor in industrial wastewater treatment.

### 2.7. Reaction Pathway of 2,4-Dinitrotoluene on Persulfate Integrated with Ag (4 wt%)/NiO under Ultrasonic Irradiation

Within the period of the tests on persulfate cooperated with Ag (4 wt%)/NiO motivated by ultrasound, most of the reaction intermediates obtained from microextraction were cautiously identified on a GC-MS spectrometer. Table 2 summarizes the outcomes, wherein the ingredients comprise 2,4-dinitrotoluene feedstock, *o*-mononitrotoluene, *p*-mononitrotoluene, nitrobenzene, 2-nitrophenol, 3-nitrophenol, 4-nitrophenol, phenol, hydroquinone, and *p*-benzoquinone. As concerns *o*-mononitrotoluene and *p*-mononitrotoluene, the electron-donating methyl group gives rise to the enhancement of the electron density of nitro groups, leading to the denitration of 2,4-dinitrotoluene [75]. However, the occurrence of nitrobenzene reveals the degradation pathway of oxidation on the methyl group of *o*-mononitrotoluene or *p*-mononitrotoluene, followed with the cleavage of carboxylic acid. With regard to 2-nitrophenol, 3-nitrophenol, and 4-nitrophenol, they could be descended from hydroxycyclohexadienyl radicals, which proceeded with the addition of O_2_ and the sequential detachment of HO_2_• into hydroxylated compounds [66,76]. The denitration of nitrophenols happened explicitly on account of the phenol sensed [77]. Obviously, hydroquinone was thought to be a reactive intermediate of phenol, which would be successively transformed into *p*-benzoquinone via hydrogen abstraction based on our previous studies [35,36]. Ultimately, 2,4-dinitrotoluene was completely decomposed into carbon dioxide, nitrate ions (sensed at UV-Vis 313 nm), and water. On the basis of the authenticated degradation compounds, Figure 11 illustrates the plausible 2,4-dinitrotoluene oxidation pathways opened up by sono-motivated persulfate integrated with Ag/NiO semiconductors.

## 3. Experimental Methods

### 3.1. Test on Persulfate Integrated with Semiconductors Motivated by Ultrasound

Experimental system composed of main apparatus was identical to that used in our previous work [35]. The sonolytic reactor was continuously imposed by ultrasound at the frequency of 20 kHz with a changeable power intensity from an ultrasonic generator equipped with a titanium probe in the dimensions of Φ13 mm × 60 mm (Chrom Technol. Corp. UP-1200, Bellows Falls, VT, USA). The reactor was made of double jacket cylinder, wherein semiconductors were packed into a basket, completely immersed in the aqueous solution. The operating temperature was held steadily at 318 K by means of a thermostat fitted with a water circulation loop [26]. Prior to the testing, some real wastewater at the 180 mg L^−1^ concentration of 2,4-dinitrotoluene accompanied with traces of sulfate and nitrate anions originated from mixed acids catalysts on the basis of GC-MS and ion chromatography analyses (rendered from military ammunition plants) was well agitated with the proportional weight of sodium persulfate (≥99.5%, Fluka, Buchs, Switzerland) and diverse semiconductors, respectively, including ZnO, Ni_2_O_3_, Fe_2_O_3_, SnO_2_ (≥99.0%, mentioned above purchased from Riedel-de Haen), NiO, PbO (≥99.9%, Alfa Aesar, Haveril, MA, USA), Cu_2_O (SHOWA), TiO_2_ (Degussa P-25), and Pb_3_O_4_ (99.9%, Sigma-Aldrich, St. Louis, MO, USA). In this research, batch-mode experiments were executed for 3 h at atmospheric pressure. For the duration of the testing, the wastewater was sampled at 30 min intervals from the reactor, immediately quenched by an ice bath (273–277 K) [78]. Subsequently, a total organic carbon (TOC) analyzer was used for the measurement of the contained organic compound content.

For the sake of verifying the influence of the band gap energy of semiconductors on the removal efficiency of 2,4-dinitrotoluene, a series of tests were carried out on Ag (1–4 wt%)/NiO, manufactured via commercial NiO powder impregnated with weighted silver nitrate (≥99.5%, Riedel-de Haen, Berlin, Germany) and calcined at 473 K for 3 h. In addition, experiments with a variety of ultrasonic power intensities (140 to 260 W cm^−2^, ultrasonic power output/titanium-tip area) were executed to elucidate the effects of ultrasonic irradiation. In order to achieve an optimal persulfate concentration on 2,4-dinitrotoluene decomposition rates, some tests were performed at persulfate concentrations between 40.0 and 70.0 mM. On the other hand, experiments with diverse Ag/NiO dosages (1.05 up to 1.50 g L^−1^) were fulfilled as well. In this study, all the experiments were accomplished at least in duplicate to confirm the data acquired.

### 3.2. Total Organic Carbon (TOC) Analysis

For the duration of the experiments on persulfate integrated with semiconductors motivated by ultrasound, the wastewater was regularly sampled and instantly analyzed using a total organic carbon instrument (GE Corp. Sievers InnovOx, Trevose, PA, USA), installed with a nondispersive infrared (NDIR) detector. All of the organic ingredients in the samples would be completely mineralized into carbon dioxide by way of sodium persulfate at the supercritical condition of water. On the contrary, the inorganic pollutants would be decomposed into carbonic acid via the acidification treatment of phosphoric acid. Original data were calibrated to the standard curve, established cautiously by the potassium hydrogen phthalate solutions between the concentrations of 0 and 500 mg L^−1^.

### 3.3. Physicochemical Characterizations of Ag/NiO 

The surface morphology and silver contents of Ag/NiO were examined by way of a precise field-emission scanning electron microscope (FE-SEM, JSM-6500F, JEOL, Tokyo, Japan) cooperated with an energy dispersive X-ray spectroscope (EDS, JED-2300, JEOL, Tokyo, Japan). The crystalline structures of original Ag/NiO were analyzed using an X-ray diffractometer (Brucker, Advance-D825A, Billerica, MA, USA) equipped with monochromated high-intensity CuKα radiation (λ = 1.5418 Å) in an accelerating voltage of 40 kV and emission current of 30 mA among the 2θ range of 10–80°. By means of a UV-Vis spectrometer (UV-DRS, Lambda 850, PerkinElmer, Waltham, MA, USA), the ultraviolet-visible diffuse reflectance spectra of Ag/NiO were determined, wherein an integrating sphere was performed at the wavelength of 380 to 800 nm, referring to the chemical compound of BaSO_4_. The surface electronic states of the original and reacted Ag/NiO were examined utilizing an X-ray photoelectron spectrometer (XPS, Axis Ultra, Kratos Analytical, Stretford, UK) installed with a monochromatic AlKα source at the radiation energy of 1486.71 eV. The C 1s core level of adventitious carbon (284.8 eV) served as the reference state of the binding energy.

### 3.4. Scavenging Effects

To elucidate the chief oxidizing agents, the mineralization of 2,4-dinitrotoluene using the process of persulfate integrated with Ag/NiO under ultrasonic irradiation was also carried out in the existence of scavengers, including benzene, methanol, and ethanol, respectively [17,71]. The 2,4-dinitrotoluene content in wastewater was detected at the wavelength of 254 nm using a UV-Vis spectrophotometer (Lambda 850, PerkinElmer, Waltham, MA, USA) [30]. Especially, the difference in the removal efficiency of 2,4-dinitrotoluen in the absence/presence of benzene served as an indirect sign to the yield of sulfate radicals [36]. Transparently, the benzene scavenging effect could be magnified with higher yields of sulfate radicals, leading to an intense difference in the 2,4-dinitrotoluene removal efficiency. Therefore, the benzene scavenging index would be reasonably substituted for sulfate radical yields upon exploring the influence of operating parameters on 2,4-dinitrotoluene removal rates. 

### 3.5. Gas Chromatograph–Mass Spectrometer Analysis (GC-MS)

The wastewater (300 mL) was withdrawn from the sonolytic reactor until the process of persulfate coupled with Ag/NiO motivated by ultrasound was conducted for 1.5 h. The typical microextraction fiber (Carboxen/Polydimethylsiloxane, 75 μm, Supelco, Bellefonte, PA, USA) was completely immersed into the aqueous solution to adsorb and concentrate simultaneously the organic compounds involved. Sequentially, the fiber was packed tightly into a syringe which was directly connected to the injection port of the gas chromatograph-mass spectrometer (Hewlett Packard 59864B/HP 5973 MASS). Analyses were carried out by a capillary column with dimensions of 30 m × 0.25 mm (film thickness 0.25 μm, UA-5, Metal ULTRA ALLOY, Fukushima, Japan), wherein helium gas (99.995%) acted as the carrier gas and the operation temperature was programmed from 303 to 573 K at a heating rate of 20 K min^−1^. Most degradation intermediates of 2,4-dinitrotoluene were explicitly identified on the basis of mass spectra referenced to those of the database (Wiley 275 L, Hoboken, NJ, USA) with authority.

## 4. Conclusions

On the basis of the above discussion, it is apparent that 2,4-dinitrotoluene pollutants can be predominantly mineralized by reactive sulfate radicals which originate from persulfate anions, motivated via either ultrasound or sono-induced electrons on semiconductors. The higher removal rate of 2,4-dinitrotoluene was achieved with semiconductors with lower band gap energy due to being easily excited by sonoluminescence. Additionally, the 2,4-dinitrotoluene degradation rate was retarded sequentially by benzene, ethanol, and methanol, indicating that sulfate radicals are the main oxidants. In accordance with GC-MS analyses, it could be reasonably hypothesized that 2,4-dinitrotoluene was initially denitrated into *o*-mononitrotoluene or *p*-mononitrotoluene, followed with decarboxylation into nitrobenzene. Then, nitrobenzene was oxidized to 2-nitrophenol, 3-nitrophenol, and 4-nitrophenol, respectively, by way of hydroxycyclohexadienyl radicals. Phenol was definitely monitored on account of the denitration of nitrophenol compounds and was sequentially decomposed into hydroquinone and *p*-benzoquinone. In this research, 2,4-dinitrotoluene could be almost completely eliminated. This means that persulfate integrated with suitable semiconductors irradiated by ultrasound is a promising method for the disposal of wastewater in toluene nitration processes.

## Figures and Tables

**Figure 1 molecules-28-04351-f001:**
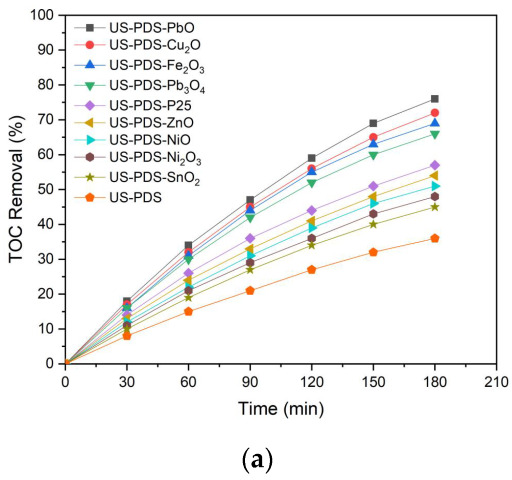
(**a**) Time-dependent drawings of TOC removal efficiency executed by the persulfate oxidation and persulfate integrated with various semiconductors under ultrasonic irradiation, respectively. (**b**) The increment in TOC removal efficiency between presence and absence of semiconductors versus band gap energy of various semiconductors. (US: ultrasonic irradiation, PDS: peroxydisulfate anions).

**Figure 2 molecules-28-04351-f002:**
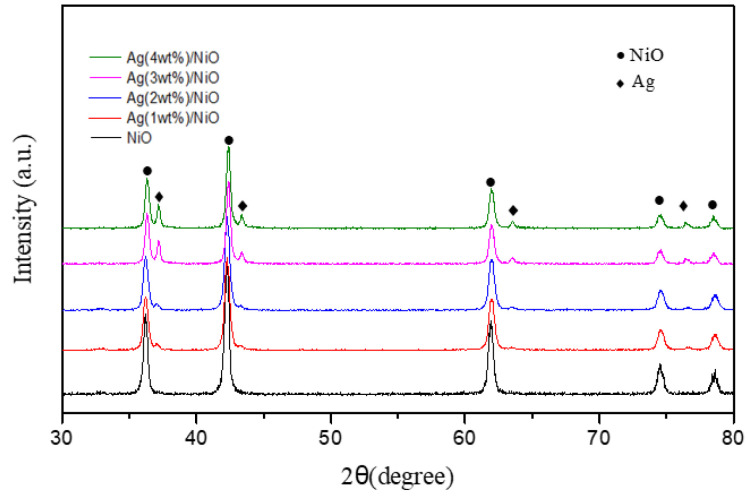
X-ray diffraction patterns of Ag (1 wt%)/NiO, Ag (2 wt%)/NiO, Ag (3 wt%)/NiO, Ag (4 wt%)/NiO, and NiO semiconductors.

**Figure 3 molecules-28-04351-f003:**
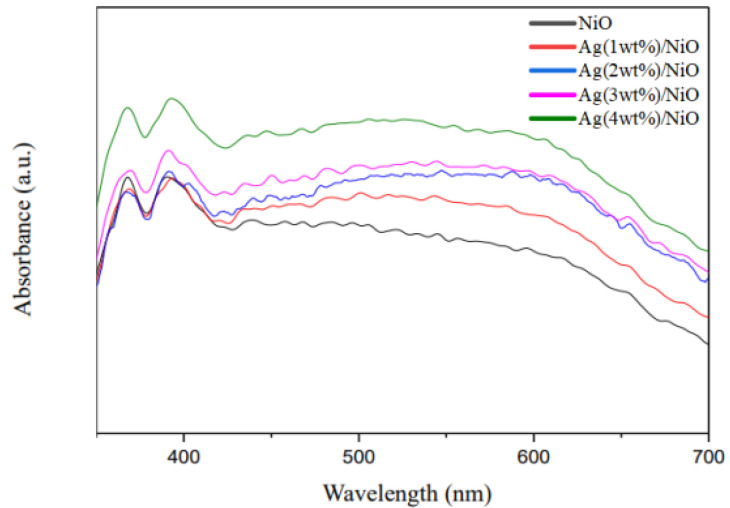
UV-vis diffuse reflectance spectra of Ag (1 wt%)/NiO, Ag (2 wt%)/NiO, Ag (3 wt%)/NiO, Ag (4 wt%)/NiO, and NiO semiconductors.

**Figure 4 molecules-28-04351-f004:**
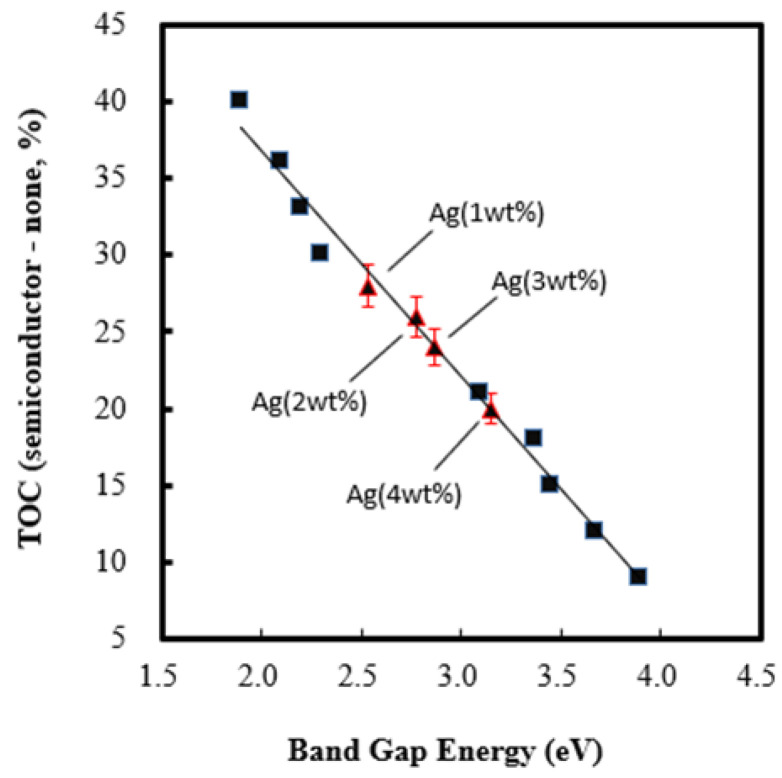
The increment in TOC removal efficiency between presence and absence of Ag (1–4 wt%)/NiO versus band gap energy of Ag (1–4 wt%)/NiO semiconductors.

**Figure 5 molecules-28-04351-f005:**
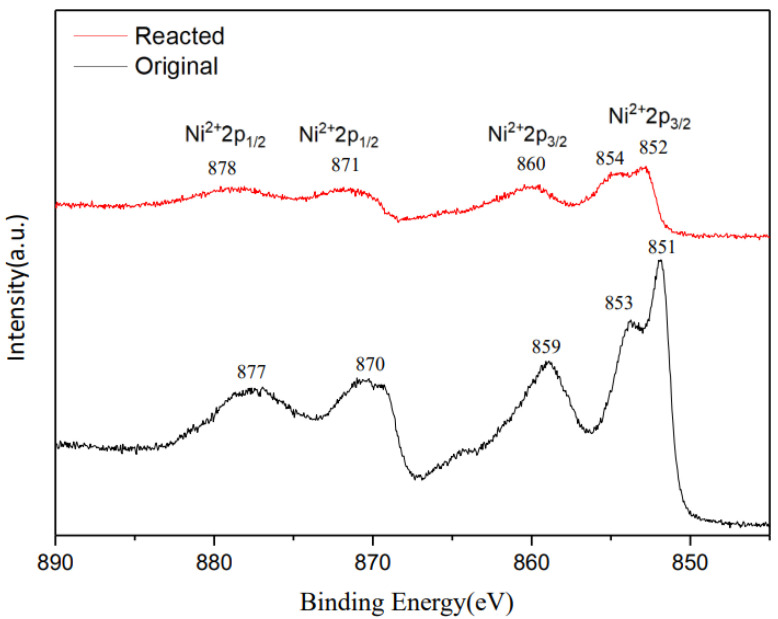
X-ray photoelectron spectra of Ni 2p core level for original Ag (4 wt%)/NiO and reacted Ag (4 wt%)/NiO semiconductors.

**Figure 6 molecules-28-04351-f006:**
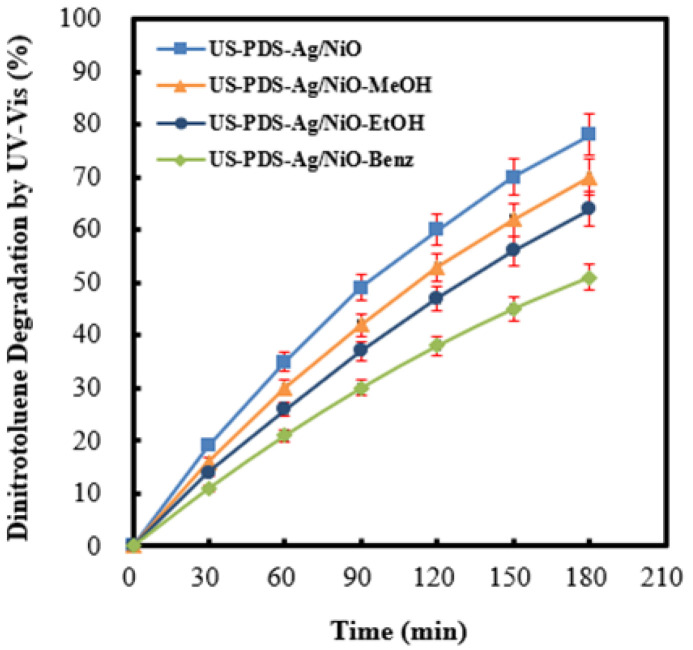
Effect of coexistence of benzene, ethanol, and methanol, respectively, on the 2,4-dinitrotoluene degradation efficiency in the process of persulfate integrated with Ag (4 wt%)/NiO motivated by ultrasound. (US: ultrasonic irradiation, PDS: peroxydisulfate anions).

**Figure 7 molecules-28-04351-f007:**
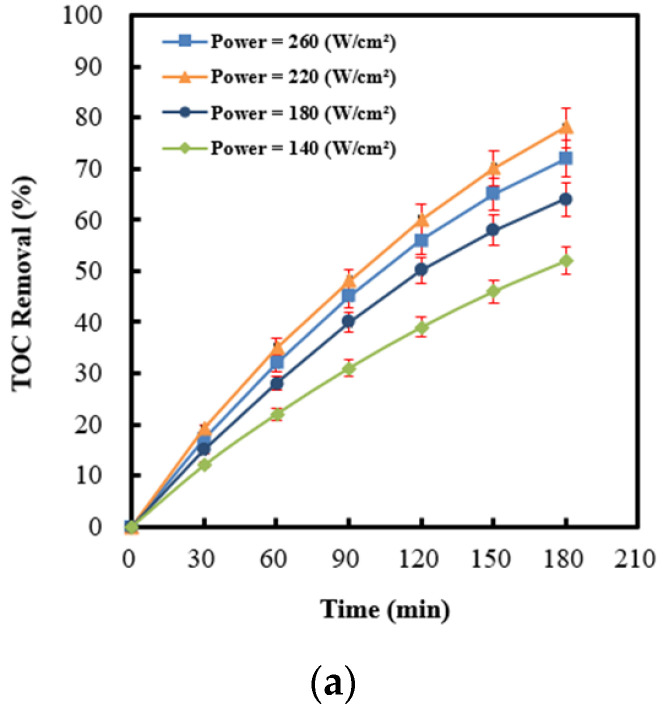
(**a**) Effect of ultrasonic power intensity on the TOC removal efficiency in the process of persulfate integrated with Ag (4 wt%)/NiO motivated by ultrasound. (**b**) The difference in 2,4-dinitrotoluene degradation efficiency between the absence and presence of benzene detected by a UV-Vis spectrophotometer, served as benzene scavenging effects.

**Figure 8 molecules-28-04351-f008:**
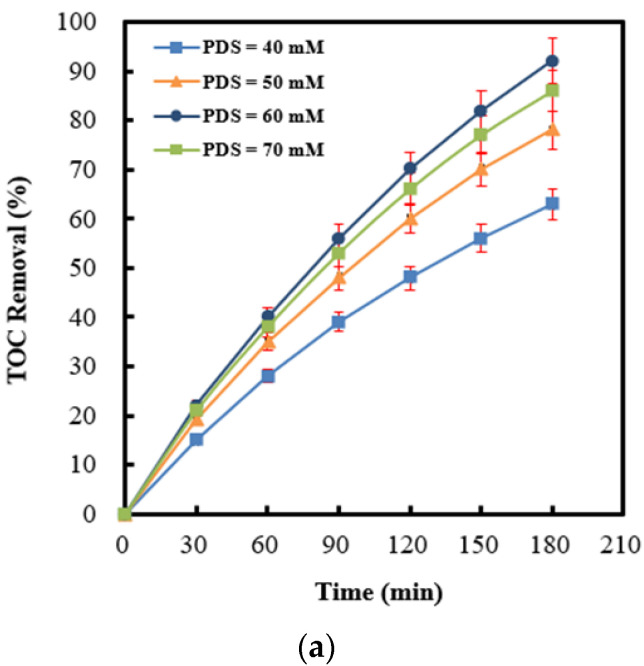
(**a**) Effect of persulfate anion concentrations on the TOC removal efficiency in the process of persulfate integrated with Ag (4 wt%)/NiO motivated by ultrasound. (**b**) The difference in 2,4-dinitrotoluene degradation efficiency between the absence and presence of benzene, detected by a UV-Vis spectrophotometer, served as benzene scavenging effects (PDS: peroxydisulfate anions).

**Figure 9 molecules-28-04351-f009:**
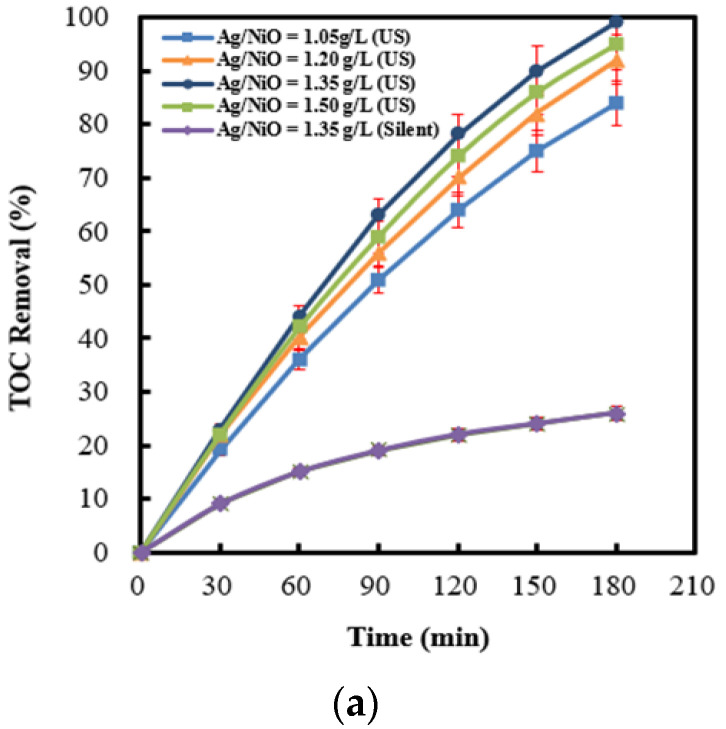
(**a**) Effect of Ag (4 wt%)/NiO dosages on the TOC removal efficiency in the process of persulfate integrated with Ag (4 wt%)/NiO motivated by ultrasound. (**b**) The difference in 2,4-dinitrotoluene degradation efficiency between the absence and presence of benzene, detected by a UV-Vis spectrophotometer, served as benzene scavenging effects. (US: ultrasonic irradiation, Silent: absence of ultrasonic irradiation).

**Figure 10 molecules-28-04351-f010:**
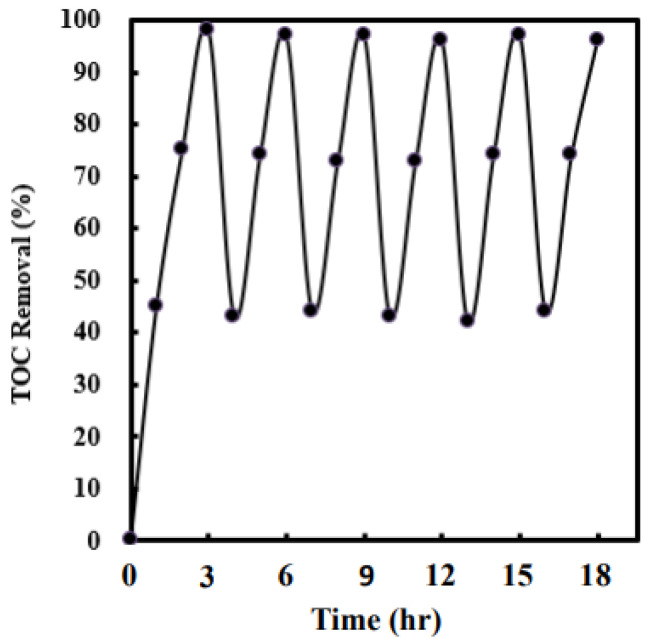
The sono-catalytic stability of Ag (4 wt%)/NiO was examined via repetitions of six tests under optimal operating conditions.

**Figure 11 molecules-28-04351-f011:**
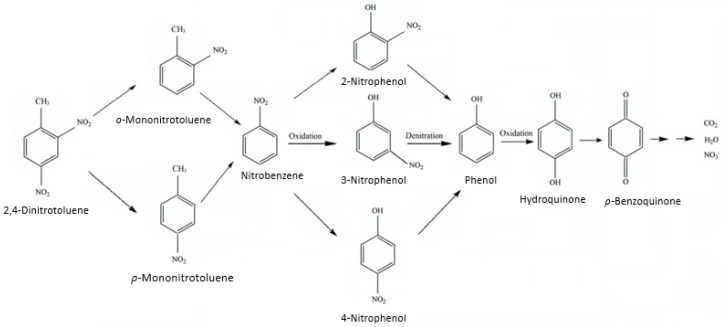
Plausible oxidation pathways of 2,4-dinitrotoluene in aqueous solution in the process of persulfate integrated with Ag/NiO motivated by ultrasound.

**Table 1 molecules-28-04351-t001:** The elemental compositions of semiconductors by EDS analyses and band gap energy by UV-DRS.

**Semiconductor**	**Ag (wt%)**	**Ni (wt%)**	**O (wt%)**	**Band Gap Energy (eV)**
NiO	0.00	86.48	13.52	3.45
Ag (1 wt%)/NiO	0.97	81.37	17.66	3.15
Ag (2 wt%)/NiO	1.43	81.11	17.46	2.87
Ag (3 wt%)/NiO	2.56	81.04	16.40	2.77
Ag (4 wt%)/NiO	3.90	78.08	18.02	2.53

**Table 2 molecules-28-04351-t002:** Compositions of feedstock and degradation intermediates determined by GC-MS.

**Component**	***m*/*z* (Relative Abundance, %)**
Feedstock	
2,4-Dinitrotoluene Degradation intermediate	51 (13.2), 63 (35.7), 78 (16.4), 89 (60.8), 90 (26.1), 119 (25.5), 165 (100), 166 (13.9)
*o*-Mononitrotoluene	39 (28.2), 63 (27.7), 65 (82.9), 77 (30.4), 89 (30.8), 91 (61.3), 92 (62.3), 120 (100)
*p*-Mononitrotoluene	39 (24.9), 63 (25.4), 65 (71.0), 77 (26.8), 89 (20.9), 91 (100), 107 (34.5), 137 (86.9)
Nitrobenzene	50 (15.6), 51 (37.6), 65 (13.5), 74 (8.9), 77 (100), 78 (7.4), 93 (16.9), 123 (70.0)
2-Nitrophenol	39 (15.6), 53 (9.5), 63 (20.1), 64 (13.9), 65 (25.4), 81 (19.5), 109 (18.0), 139 (100)
3-Nitrophenol	39 (35.8), 53 (10.6), 63 (14.7), 64 (7.9), 65 (63.7), 81 (15.8), 93 (51.3), 139 (100)
4-Nitrophenol	39 (44.2), 53 (23.2), 63 (287.1), 65 (79.9), 81 (32.9), 93 (27.0), 109 (67.0), 139 (100)
Phenol	38 (5.2), 39 (12.5), 40 (6.9), 55 (6.3), 63 (6.5), 65 (20.8), 66 (27.2), 94 (100), 95 (7.6)
Hydroquinone	39 (6.9), 53 (14.3), 54 (12.8), 55 (10.5), 81 (25.4), 82 (12.2), 110 (100), 143 (9.6)
*p*-Benzoquinone	26 (18.0), 52 (17.7), 53 (17.2), 54 (63.2), 80 (28.3), 82 (36.0), 108 (100), 110 (12.0)

## Data Availability

Data is unavailable due to privacy and ethical restrictions.

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
