# Peer review of "Ultrasound-Assisted Mineralization of 2,4-Dinitrotoluene in Industrial Wastewater Using Persulfate Coupled with Semiconductors"

_molecules, 2023, doi:10.3390/molecules28114351_

Round 1

Reviewer 1 Report

The results presented on the decomposition of 2,4-dinitrotouene in wastewater by persulfate is an interesting study and in my opinion worth publishing. However, some remarks presented below should be considered by the Authors.

 Chapters 2.3 and 3.2 have the same title. It should be changed.

 Page 3, lines 1-5 of the bottom. The Authors have stated “To elucidate the chief oxidizing agents, mineralization of 2,4-dinitrotoluene using the process of persulfate integrated with Ag/NiO under ultrasonic irradiation was also carried out in the existence of scavengers, including benzene, methanol and ethanol respectively [14, 44]. The 2,4-dinitrotoluene content in wastewater was detected at the wavelength of 254 nm by an UV-Vis spectrophotometer (Lambda 850, PerkinElmer) [27].” Do the absorption of benzene, methanol. and ethanol not interfere with that of 2,4-dinitrotoluene at 254 nm? The presentation of UV-vis spectra of these compounds in the Supplementary Material would be helpful.

 The abbreviations used in Figures 1a, 5, 7a, and 8a should be explained.

Figure 3 contains the data presented in Figure 2. In my opinion, Fig.2 should be omitted.

 The text was basically understandable to me. I am not an English native speaker and I hesitate to make language remarks, but I feel that some corrections should be made. For example, the phrases used “chief oxidant” and “motivated by ultrasound”, which are repeated throughout the text, sound unprofessionally to me. Therefore, I strongly recommend that the English usage be thoroughly revised.

Author Response

Reviewer 1:

The results presented on the decomposition of 2,4-dinitrotouene in wastewater by persulfate is an interesting study and in my opinion worth publishing. However, some remarks presented below should be considered by the Authors.

Thank you very much for your kind instructions. According to the questionnaire listed, the revision and/or response are as follows:

  1. Chapters 2.3 and 3.2 have the same title. It should be changed.

Ans: The title of Section 3.2 has been rewritten as reviewer instructed. (Page 4, 3.2. Effect of band gap energy on persulfate integrated with Ag/NiO semiconductor)

  1. Page 3, lines 1-5 of the bottom. The Authors have stated “To elucidate the chief oxidizing agents, mineralization of 2,4-dinitrotoluene using the process of persulfate integrated with Ag/NiO under ultrasonic irradiation was also carried out in the existence of scavengers, including benzene, methanol and ethanol respectively [14, 44]. The 2,4-dinitrotoluene content in wastewater was detected at the wavelength of 254 nm by an UV-Vis spectrophotometer (Lambda 850, PerkinElmer) [27].” Do the absorption of benzene, methanol. and ethanol not interfere with that of 2,4-dinitrotoluene at 254 nm? The presentation of UV-vis spectra of these compounds in the Supplementary Material would be helpful.

Ans: (a)The absorption peak of benzene in the UV-Vis spectrum is observed at the wavelength of 205 nm. (referred to T. Soltani, B.-K. Lee, “Novel and facile synthesis of Ba-doped BiFeO3 nanoparticles and enhancement of their magnetic and photocatalytic activities for complete degradation of benzene in aqueous solution” J. Hazard. Mater. 316 (2016) 122-133.) (b) The absorption peak of ethanol in the UV-Vis spectrum is observed at the wavelength of 420 nm. Besides, the absorption peak of methanol in the UV-Vis spectrum is observed at the wavelength of 415 nm. (referred to P. Shah, B. Dev, A. Deo, B. Neupane, A. Bhattarai, “UV-VIS investigation of methyl red in presence of sodium dodecyl sulfate/methanol/ethanol/water system” J. Molecular Liquids 349 (2022) 118119.) Therefore, the scavengers would not interfere with the absorption peak of 2,4-dinitrotoluene.

  1. The abbreviations used in Figures 1a, 5, 7a, and 8a should be explained.

Ans: The abbreviations used in Figures 1a, 5, 7a, and 8a have been annotated as reviewer instructed. (Page 5, US: Ultrasonic irradiation, PDS: Peroxydisulfate anions)

  1. Figure 3 contains the data presented in Figure 2. In my opinion, Fig.2 should be omitted.

Ans: In this work, the band gap energy of Ag/NiO was obtained from the spectra of UV-DRS in Fig. 2, wherein the data have been scarcely found in the literature. Nonetheless, Fig. 3 presents the correlation between the increment on TOC removal efficiency and band gap energy. Therefore, the authors ask for retention of Fig. 2 in this manuscript. 

Reviewer 2 Report

The manuscript evaluates the oxidative degradation of 2, 4 dinitrotoluene under aqueous conditions using persulfate components in combination with low band gap semiconductor materials. The manuscript largely delivers the message clearly, but there are several concerns which should be clarified for better understanding among targeted readers.

The questions/comments/suggestions are as give below:

1. On page 2, equation (4), please explain the meaning of ")))". 

2. Since this reaction is happening in aqueous conditions and the targeted molecules may be showing very low solubility in water then how did authors ensured that the reaction in occurring homogeneously in the aqueous mixture?

3. (a) Did authors observe any phase separation in the reaction mixture during/after the reaction? (b) Does author believe that the oxidative degradation might be happening at the interface? If yes, please justify with evidence. In case no, please explain the same.

4. I suggest authors to include the error bars for the data to ensure the experimental reproducibility and helpful for the readers to understand the scope of standard deviation in the data.

5. If possible, kindly share after how many cycles authors need to replace/regenerate the semiconductor basket while conducting the oxidative degradation of 2,4 dinitrotoluene. 

I suggest authors to proof read the manuscript carefully to avoid spelling/grammatical errors. 

Author Response

Reviewer 2:

The manuscript evaluates the oxidative degradation of 2, 4 dinitrotoluene under aqueous conditions using persulfate components in combination with low band gap semiconductor materials. The manuscript largely delivers the message clearly, but there are several concerns which should be clarified for better understanding among targeted readers.

Thank you very much for your kind instructions. According to the questionnaire listed, the revision and/or response are as follows:

  1. On page 2, equation (4), please explain the meaning of ")))". 

Ans: Equation (4) has been rewritten as reviewer instructed. (Page 2, S2O82- + US → 2SO4· Wherein US denotes ultrasonic irradiation )

  1. Since this reaction is happening in aqueous conditions and the targeted molecules may be showing very low solubility in water then how did authors ensured that the reaction in occurring homogeneously in the aqueous mixture?

Ans: The real wastewater was rendered from military ammunition plants. The aqueous solution was clear rather than cloudy during the testing. Therefore, oxidative degradation of 2,4-dinitrotoluene occurred homogeneously.  

  1. (a) Did authors observe any phase separation in the reaction mixture during/after the reaction? (b) Does author believe that the oxidative degradation might be happening at the interface? If yes, please justify with evidence. In case no, please explain the same.

Ans: (a) No phase separation was observed during the testing. (b) The solubility of 2,4-dinitrotoluene in neutral aqueous solution is about 200 mg/L. The solubility of 2,4-dinitrotoluene increases with an increase of acidity of solution. In this work, the 2,4-dinitrotoluene concentration of feedstock was about 182 mg/L, which is less than solubility. Therefore, no phase separation was observed.

  1. I suggest authors to include the error bars for the data to ensure the experimental reproducibility and helpful for the readers to understand the scope of standard deviation in the data.

Ans: In this work, all experiments were accomplished at least in duplicate to confirm the data reliable. The difference among same tests was less than 4%. Besides, there are many data points shown in the figures, such as Fig. 1. It is difficult to make comparison among data points by means of error bars. Therefore, the authors ask for original type of figures reserved.

  1. If possible, kindly share after how many cycles authors need to replace/regenerate the semiconductor basket while conducting the oxidative degradation of 2,4 dinitrotoluene. 

Ans: The sono-catalytic stability of Ag(4 wt%)/NiO was examined via repetitions of six tests (shown in Fig. 9). 

Reviewer 3 Report

The article reports on the oxidative degradation of 2,4-dinitrotoluenes in an aqueous solution using per-sulfate combined with semiconductors motivated by ultrasound. The authors investigated the effects of various operation variables on the sono-catalytic performance using synthesized  Ag/NiO with different weight fractions of Ag, including ultrasonic power intensity, the dosage of persulfate anions, and semiconductors. The authors hypothesized that the chief oxidants were sulfate radicals originating from persulfate anions, motivated via either ultrasound or sono-catalysis of semiconductors.

In my opinion, the structure of the manuscript is accepted and its English and written way are good, however, the results and their discussion are not presented in an acceptable manner. Therefore,  before it can be accepted for publication, major revisions are necessary; the main points to be addressed are:

 1.       The authors should provide a brief description of the sonochemical process in the introduction.

 2.       In lines 45-46, the authors mentioned “…..wherein hydroxyl radicals were considered to be principal oxidants.”. the authors should mention also that the holes, as well, are considered principal oxidants. I encourage the authors to check the following references and cite them in the text: https://doi.org/10.1016/j.cattod.2021.07.013 and https://dx.doi.org/10.1021/acsaem.0c00826.

3.       The final sentence in the first paragraph of the introduction does not seem to be connected well with this paragraph. The reviewer suggested that the authors say that this compound, as well as other organic pollutants, is harmful to the environment and human health. They can mention several methods that can be used to remove organic pollutants from water, such as… Then, the authors can mention their final sentence in lines 33-34.  they can refer to this review (https://doi.org/10.3390/catal11030317) regarding this point.

4.       Equations (1-4) in the introduction were not referring in the text.

5.       What the abbreviation “Men+” in equation 2 belongs to?

6.       The authors stated in lines 194-195 that an increase in doping amounts of Ag resulted in the observation of more clumps of particulates. They concluded that Ag metal was well dispersed on the surface of NiO, but this hypothesis lacks scientific evidence. Therefore, including TEM images inside the manuscript and EDX images in the supporting file, is necessary to substantiate their conclusion.

7.       The authors assumed that the wastewater contained only minor inorganic pollutants, how they assumed that? Lines 126.

8.       I encouraged the authors to provide a figure containing a diagram of the energy levels (CB and VB) for the tested catalysts.

9.       The UV-vis spectra of pristine NiO and loaded Ag(1wt%)/NiO (Figure 2) should be converted to their corresponding Tauc plots (authors check the following reference https://doi.org/10.3390/catal11010107 and cite it in the text) to show the effect of Ag loading on the optical bandgap.

10.   Given that the authors utilized wastewater for their catalytic experiments, it is reasonable to assume that the TOC value was high. As shown in Figure 1a, the removal efficiency was approximately 75%. However, it is important to clarify whether this value pertains to all the contents of the wastewater. The authors should provide the TOC value before conducting the experiments and briefly explain the composition of the organic compounds and ions present, as they may impact the reaction positively or negatively.

11.   Throughout the manuscript, the authors referred to the Ag metal as a dopant, but it is more likely that it is loaded onto the surface of NiO rather than incubated within it.

12.   Since the authors provided XPS analysis in Figure 4 to demonstrate the conversion of Ni+2 to Ni+3, it would be beneficial if they could also perform the same analysis to determine the nature of Ag (i.e. Ag metal or Ag+ oxide).

13.   Did the authors measure the pH before and after the catalytic experiments, especially for the experiments in sections 3.4 and 3.5?

14.   XRD analysis is required for the synthesis materials, which I suppose will show the pattern of Ag metal with NiO.

15.   In section 3.3, line 259, the authors explored the influence of scavengers such as benzene, ethanol, and methanol on the degradation rate of 2,4-dinitrotoluene. To eliminate the possible contribution of other active species in the catalytic process, such as superoxide radical, hydroxyl radical, that could be produced during the activation of the catalyst, it is highly recommended to conduct additional scavenger experiments using benzoquinone, potassium iodide, and 2-methylpropan-2-ol. The authors can refer to the following studies (https://doi.org/10.1016/j.jpcs.2021.110287, https://dx.doi.org/10.1021/acscatal.0c01713, https://doi.org/10.3390/catal11121423) and cite them in the text.

16.   Furthermore, based on equations (5 to 7) in lines 184-186, the authors can potentially identify the primary role of holes and/or electrons in the generation of SO4 radicals by analyzing the outcomes of these scavenger experiments.

17.   Based on the Gc-Ms analysis shown in Table 2, I suppose it will be difficult to identify some of the mentioned compounds without derivatization. How can the authors explain the identification of the phenolic compounds without derivatization?!

check the report

Author Response

Reviewer 3:

The article reports on the oxidative degradation of 2,4-dinitrotoluenes in an aqueous solution using per-sulfate combined with semiconductors motivated by ultrasound. The authors investigated the effects of various operation variables on the sono-catalytic performance using synthesized  Ag/NiO with different weight fractions of Ag, including ultrasonic power intensity, the dosage of persulfate anions, and semiconductors. The authors hypothesized that the chief oxidants were sulfate radicals originating from persulfate anions, motivated via either ultrasound or sono-catalysis of semiconductors.

In my opinion, the structure of the manuscript is accepted and its English and written way are good, however, the results and their discussion are not presented in an acceptable manner. Therefore,  before it can be accepted for publication, major revisions are necessary; the main points to be addressed are:

Thank you very much for your kind instructions. According to the questionnaire listed, the revision and/or response are as follows:

  1. The authors should provide a brief description of the sonochemical process in the introduction.

Ans: The sonochemical process for treatment of 2,4-dinitrotoluene has been issued in the introduction section at second paragraph: (Page 1, Up to the present, several publications have focused on the sonochemical decomposition of 2,4-dinitrotoluene [5-7]. Thermal pyrolysis is proposed to be main degradation mechanism on 2,4-dinitrotoluene oxidation, wherein sonolytic temperature is the most significant influencing factor. Sonochemical destruction of 2,4-dinitrotoluene could be obviously accelerated with assistance of TiO2 powder, which supplied extra nuclei for generation of cavitation microbubbles.)

  1. In lines 45-46, the authors mentioned “…..wherein hydroxyl radicals were considered to be principal oxidants.”. the authors should mention also that the holes, as well, are considered principal oxidants. I encourage the authors to check the following references and cite them in the text: https://doi.org/10.1016/j.cattod.2021.07.013 and https://dx.doi.org/10.1021/acsaem.0c00826.

 Ans: The sentences have been rewritten as reviewer instructed and references have been cited. (Page 2, wherein hydroxyl radicals and holes were considered to be principal oxidants [15, 16].)

  1. The final sentence in the first paragraph of the introduction does not seem to be connected well with this paragraph. The reviewer suggested that the authors say that this compound, as well as other organic pollutants, is harmful to the environment and human health. They can mention several methods that can be used to remove organic pollutants from water, such as… Then, the authors can mention their final sentence in lines 33-34.  they can refer to this review (https://doi.org/10.3390/catal11030317) regarding this point.

Ans: The sentences have been rewritten as reviewer instructed and the reference has been cited. (Page1, There is an increasing trend for related wastes generated from activities of these industries [4].)

  1. Equations (1-4) in the introduction were not referring in the text.

Ans: In this work, the authors want to provide a novel method for activation of persulfate anions, wherein sono-catalysis of semiconductors is tested and proved. Therefore, the conventional methods for activation of persulfate (Eq.1-4) need to be listed for comparison.

  1. What the abbreviation “Men+” in equation 2 belongs to?

Ans: Equation (2) has been rewritten as reviewer instructed. (Page 2, S2O82- + Metaln+ → SO4·- +SO42- + Metal(n+1)+ )

  1. The authors stated in lines 194-195 that an increase in doping amounts of Ag resulted in the observation of more clumps of particulates. They concluded that Ag metal was well dispersed on the surface of NiO, but this hypothesis lacks scientific evidence. Therefore, including TEM images inside the manuscript and EDX images in the supporting file, is necessary to substantiate their conclusion.

Ans: The sentences have been rewritten as reviewer instructed. The Ag metal content was estimated by means of EDS element analyses. (Page 5, Thus, Ag metal appears to be well deposited on the surface of NiO.)

  1. The authors assumed that the wastewater contained only minor inorganic pollutants, how they assumed that? Lines 126.

Ans: The sentences have been rewritten as reviewer instructed. (Page 3, On the contrary, the inorganic pollutants would be decomposed into carbonic acid via acidification treatment of phosphoric acid.)

  1. I encouraged the authors to provide a figure containing a diagram of the energy levels (CB and VB) for the tested catalysts.

Ans: In this work, the increment of TOC removal efficiency was correlated with band gap energy of semiconductors no matter with their species. Ag/NiO was prepared to supply semiconductors with different band gap energy. Therefore, the energy levels of Ag/NiO were neglected in the manuscript.

  1. The UV-vis spectra of pristine NiO and loaded Ag(1wt%)/NiO (Figure 2) should be converted to their corresponding Tauc plots (authors check the following reference https://doi.org/10.3390/catal11010107 and cite it in the text) to show the effect of Ag loading on the optical bandgap.

Ans: In this work, the band gap energy of Ag/NiO was evaluated on the basis of Tauc’s relation [(αhn)1/n = A(hn-Eg)], wherein the estimated method has been described in detail in the text. The references suggested by reviewer hves been cited in this manuscript. (Page 6, the band gap energy was estimated from the intercept of the tangent line to the X-axis [60-63]. Accordingly, the band gap energy of Ag(1-4 wt%)/NiO was determined between 3.15 and 2.53 eV (refer to Table 1), consistent with results of Pandey et al. [64].)

  1. Given that the authors utilized wastewater for their catalytic experiments, it is reasonable to assume that the TOC value was high. As shown in Figure 1a, the removal efficiency was approximately 75%. However, it is important to clarify whether this value pertains to all the contents of the wastewater. The authors should provide the TOC value before conducting the experiments and briefly explain the composition of the organic compounds and ions present, as they may impact the reaction positively or negatively.

Ans: The 2,4-dinitrotoluene concentration in real wastewater (rendered by military ammunition plants) has mentioned as 1.0 mM in the experimental section. In fact, 2,4-dinitrotoluene was manufactured by H2SO4-HNO3 mixed acids in toluene nitration process. The compositions of organic compounds in wastewater feedstock have been identified by GC-MS (refer to Table 2).

  1. Throughout the manuscript, the authors referred to the Ag metal as a dopant, but it is more likely that it is loaded onto the surface of NiO rather than incubated within it.

Ans: It seems that Ag metal was deposited on the surface of NiO. The sentences have been rewritten as reviewer instructed. (Page 5, It indicates that most surface of NiO was smooth, coated with some irregularly shaped particulates. As doping amounts of Ag increased, more clumps of particulates were observed. Thus, Ag metal appears to be well deposited on the surface of NiO.)

  1. Since the authors provided XPS analysis in Figure 4 to demonstrate the conversion of Ni+2to Ni+3, it would be beneficial if they could also perform the same analysis to determine the nature of Ag (i.e. Ag metal or Ag+ oxide).

Ans: The XPS analyses of Ag metal make insignificant difference for comparison of original and reacted Ag/NiO.

  1. Did the authors measure the pH before and after the catalytic experiments, especially for the experiments in sections 3.4 and 3.5?

Ans: The pH of solution has been gradually changed to 4 during the experiment, due to following reactions occurred.

S2O82- + H2O → H2SO5 + SO42-

  1. XRD analysis is required for the synthesis materials, which I suppose will show the pattern of Ag metal with NiO.

Ans: The XRD patterns of Ag/NiO have been shown in Fig. S2 as reviewer instructed. (Page 5, Figure S2 illustrates X-ray diffraction patterns of Ag(1-4wt%)/NiO semiconductors. It provides another evidence for the existence of Ag metal on the surface of NiO.)

  1. In section 3.3, line 259, the authors explored the influence of scavengers such as benzene, ethanol, and methanol on the degradation rate of 2,4-dinitrotoluene. To eliminate the possible contribution of other active species in the catalytic process, such as superoxide radical, hydroxyl radical, that could be produced during the activation of the catalyst, it is highly recommended to conduct additional scavenger experiments using benzoquinone, potassium iodide, and 2-methylpropan-2-ol. The authors can refer to the following studies (https://doi.org/10.1016/j.jpcs.2021.110287, https://dx.doi.org/10.1021/acscatal.0c01713, https://doi.org/10.3390/catal11121423) and cite them in the text.

Ans: The valuable information suggested by reviewer is greatly appreciated. The reference mentioned has been cited as [64] in this manuscript. The scavenging tests using benzoquinone, potassium iodide, and 2-methylpropan-2-ol would be performed in future work.  

  1. Furthermore, based on equations (5 to 7) in lines 184-186, the authors can potentially identify the primary role of holes and/or electrons in the generation of SO4radicals by analyzing the outcomes of these scavenger experiments.

Ans: The valuable information suggested by reviewer is greatly appreciated. The hole scavenging tests would be performed in future to evaluate contributions for generation of sulfate radicals by sono-catalysis of semiconductors. 

  1. Based on the GC-MS analysis shown in Table 2, I suppose it will be difficult to identify some of the mentioned compounds without derivatization. How can the authors explain the identification of the phenolic compounds without derivatization?

Ans: Based on our previous studies in the field of oxidative degradation of nitrotoluene, nitrobenzene and aniline by means of persulafte or Fenton process, the plausible reaction pathway was proposed.  

Reviewer 4 Report

This is a very interesting manuscript presenting a new potential way to remove organic pollutants from the environment.

The work is done very carefully. All procedures are given in detail. The results are thoroughly discussed. The manuscript is well organized and well written based on well-chosen literature.

I think it would be also very interesting if the Authors gave their point of view on introducing this method to industrial applications.

Fine

Author Response

Reviewer 4:

This is a very interesting manuscript presenting a new potential way to remove organic pollutants from the environment.

The work is done very carefully. All procedures are given in detail. The results are thoroughly discussed. The manuscript is well organized and well written based on well-chosen literature.

I think it would be also very interesting if the Authors gave their point of view on introducing this method to industrial applications.

Thank you very much for your kind instructions.

Round 2

Reviewer 2 Report

Authors made an attempt to respond to the questions and concerns of the reviewer's. However, the reply is not satisfactory. For instance, in response to the reviewer's 2 questions:

1. With reference to Q#1, in Equation 4, authors corrected ")))", then why it is still  present in Equation 5. Please revise the manuscript with seriousness to rectify (at least the similar) mistakes. 

2. In Q#2, authors stated that the real waste water was rendered from ammunition plant. Please state all other impurities that waste water was containing. In absence of this information, the comparison is not equivalent to lab studies. 

3. In Q#3, author state that experiments were performed within the solubility limit of 2, 4 dinitrotoluene. (a) Why this limit was selected? Further, authors mention that the solubility of 2, 4 dinitrotoluene increases with the acidity of the solution. In this regard, on Page 12 of the manuscript mentions that the occurrence of nitrobenzene manifests the degradation pathway of oxidation on methyl group of o-mononitrotoluene or p-mononitrotoluene, followed with cleavage of carboxylic acid. (b) Where is the carboxylic acid group in the 2, 4 dinitrotoluene for enabling the cleavage?

4. It is not understood, why error bars can't be placed in Figures? How did authors reached the 4% error limit. Is it for all the different measurements in the article or only associated up to Figure 1. If error bars in Figure 1 seems too crowded, please state the error percent in that experiment but in other Figures 1(b), 3, 6 etc., I still recommend authors to include the error bars.

I suggest authors to avoid repetitive typo errors and go through the spelling check. Thank you.

Author Response

Reviewer 2:

Authors made an attempt to respond to the questions and concerns of the reviewer's. However, the reply is not satisfactory. For instance, in response to the reviewer's 2 questions

Thank you very much for your kind instructions again. According to the questionnaire listed, the revision and/or response are as follows:

  1. With reference to Q#1, in Equation 4, authors corrected ")))", then why it is still present in Equation 5. Please revise the manuscript with seriousness to rectify (at least the similar) mistakes. 

Ans: We sincerely apologize to reviewer 2 for our mistakes. The symbol ")))" in Eq. 5 has been corrected as reviewer instructed. (Page 3: Semiconductor + US ® h+vb + e-cb     )

  1. In Q#2, authors stated that the real waste water was rendered from ammunition plant. Please state all other impurities that waste water was containing. In absence of this information, the comparison is not equivalent to lab studies. 

Ans: Based on GC-MS and Ion Chromatography analyses, the main composition of real wastewater was 2,4-dinitrotoluene, accompanied with trace of sulfate and nitrate anions caused by mixed acid catalysts. There is not any other organic compound detected in wastewater. The 2,4-dinitrotoluene concentration is 180 mg/L. (Page 12: Prior to the testing, some real wastewater at the 180 mg L-1 concentration of 2,4-dinitrotoluene accompanied with traces of sulfate and nitrate anions, originated from mixed acids catalysts, (rendered from military ammunition plants))

  1. In Q#3, author state that experiments were performed within the solubility limit of 2, 4 dinitrotoluene. (a) Why this limit was selected? Further, authors mention that the solubility of 2, 4 dinitrotoluene increases with the acidity of the solution. In this regard, on Page 12 of the manuscript mentions that the occurrence of nitrobenzene manifests the degradation pathway of oxidation on methyl group of o-mononitrotoluene or p-mononitrotoluene, followed with cleavage of carboxylic acid. (b) Where is the carboxylic acid group in the 2, 4 dinitrotoluene for enabling the cleavage?

Ans: (a) Experiments were conducted below the solubility of 2,4-dinitrotoluene. It means that no phase separation occurred during the testing. (b) The sentence has been rewritten as reviewer instructed. (Page 11, However, the occurrence of nitrobenzene reveals the degradation pathway of oxidation on methyl group of o-mononitrotoluene or p-mononitrotoluene, followed with cleavage of carboxylic acid.) Based on our previous work [35, 36], the plausible reaction pathway was proposed.

  1. It is not understood, why error bars can't be placed in Figures? How did authors reached the 4% error limit. Is it for all the different measurements in the article or only associated up to Figure 1. If error bars in Figure 1 seems too crowded, please state the error percent in that experiment but in other Figures 1(b), 3, 6 etc., I still recommend authors to include the error bars.

Ans: As reviewer instructed, Figure 1(b), 4, 6, 7, 8, 9 have been corrected.

Reviewer 3 Report

I would like to thank the authors for responding to the reviewer's comments and improving their manuscript. However, I would like to point out that their responses do not fully address some of the reviewer's concerns. The primary issues that need to be addressed are as follows:

1.       Regarding comment 4, the equations (1-4) in the introduction are not referred to in the text. I suggest that the authors mention these equations in the manuscript, as follows: "The sulfate radicals could be generated via persulfate anions motivated by various manners, such as thermal energy, ultraviolet irradiation (Eq. 1), electron donation of transition metal ions (Eq. 2) [19−21], electrochemical reduction by cathode (Eq. 3) [22], and ultrasonic irradiation (Eq. 4) [23]."

2.       With reference to comment 6, if the authors cannot provide a TEM image, they should remove the following sentence from the manuscript because this hypothesis lacks scientific evidence: "Thus, Ag metal appears to be well deposited on the surface of NiO."

 3.       Regarding comment 7, the authors’ answers did not fully address the reviewer's comment. The reviewer asked mainly about the real value of TOC in the wastewater used in the experiments. Did the authors use a real wastewater? If yes, what is the value of the TOC? Because in real wastewater, there are other organic and inorganic compounds present in the sample during the measurement of the TOC.

 4.       With reference to comment 9, since the authors evaluated the band gap energy of Ag/NiO based on Tauc's relation, they should provide the suggested graph.

 5.       Regarding comment 10, if the authors used real wastewater, the reaction medium should have contained other compounds besides 2,4-dinitrotoluene, which must participate in the reaction, but this was not shown in the manuscript. Table 2 shows the intermediates that formed only from the degradation of 2,4-dinitrotoluene.

 6.       With reference to comment 11, since the Ag is deposited on the surface, the word "doped" must be changed throughout the manuscript to "loaded on the surface."

 7.       Regarding comment 12, the main purpose of the XPS analysis is to show the nature of Ag after preparation, not after the reaction, because it could be that the authors prepared Ag oxide instead of Ag metal.

 8.       Regarding comment 14, the authors should show the XRD analysis in the manuscript and include it in the discussion. Also, they should show in the graph the planes of Ag nanoparticles, which will confirm the formation of Ag nanoparticles. The diffraction peaks of Ag confirm the formation of the Ag nanoparticles, which can be used instead of the XPS analysis. Please refer to the following article and cite it in the text: "https://doi.org/10.3390/pharmaceutics14051104".

 9.       Regarding comment 17, the authors should mention in the manuscript that this reaction intermediate is expected based on previous studies.

The English language is accepted.

Author Response

Reviewer 3:

 I would like to thank the authors for responding to the reviewer's comments and improving their manuscript. However, I would like to point out that their responses do not fully address some of the reviewer's concerns. The primary issues that need to be addressed are as follows:

Thank you very much for your kind instructions again. According to the questionnaire listed, the revision and/or response are as follows:

  1. Regarding comment 4, the equations (1-4) in the introduction are not referred to in the text. I suggest that the authors mention these equations in the manuscript, as follows: "The sulfate radicals could be generated via persulfate anions motivated by various manners, such as thermal energy, ultraviolet irradiation (Eq. 1), electron donation of transition metal ions (Eq. 2) [19−21], electrochemical reduction by cathode (Eq. 3) [22], and ultrasonic irradiation (Eq. 4) [23]." 

Ans: This paragraph has been rewritten as reviewer instructed. (Page 2, The sulfate radicals could be generated via persulfate anions motivated by various manners, such as thermal energy, ultraviolet irradiation (Eq. 1), electron donation of transition metal ions (Eq. 2) [19-21], electrochemical reduction by cathode (Eq. 3) [22], and ultrasonic irradiation (Eq. 4) [23]. )

  1. With reference to comment 6, if the authors cannot provide a TEM image, they should remove the following sentence from the manuscript because this hypothesis lacks scientific evidence: "Thus, Ag metal appears to be well deposited on the surface of NiO."

Ans: This sentence has been removed as reviewer instructed.  

  1. Regarding comment 7, the authors’ answers did not fully address the reviewer's comment. The reviewer asked mainly about the real value of TOC in the wastewater used in the experiments. Did the authors use a real wastewater? If yes, what is the value of the TOC? Because in real wastewater, there are other organic and inorganic compounds present in the sample during the measurement of the TOC.

Ans: Based on GC-MS and Ion Chromatography analyses, the main composition of real wastewater was 2,4-dinitrotoluene, accompanied with trace of sulfate and nitrate anions caused by mixed acid catalysts. There is not any other organic compound detected in wastewater. The 2,4-dinitrotoluene concentration is 180 mg/L. (Page 12: Prior to the testing, some real wastewater at the 180 mg L-1 concentration of 2,4-dinitrotoluene accompanied with traces of sulfate and nitrate anions, originated from mixed acids catalysts, (rendered from military ammunition plants))

  1. With reference to comment 9, since the authors evaluated the band gap energy of Ag/NiO based on Tauc's relation, they should provide the suggested graph.

Ans: The band gap energy of Ag/NiO was evaluated based on Tauc's relation, wherein the graph was provided in Figure S2.

  1. Regarding comment 10, if the authors used real wastewater, the reaction medium should have contained other compounds besides 2,4-dinitrotoluene, which must participate in the reaction, but this was not shown in the manuscript. Table 2 shows the intermediates that formed only from the degradation of 2,4-dinitrotoluene.

Ans: Based on GC-MS and Ion Chromatography analyses, the main composition of real wastewater was 2,4-dinitrotoluene, accompanied with trace of sulfate and nitrate anions caused by mixed acid catalysts. There is not any other organic compound detected in wastewater. The 2,4-dinitrotoluene concentration is 180 mg/L. (Page 12: Prior to the testing, some real wastewater at the 180 mg L-1 concentration of 2,4-dinitrotoluene accompanied with traces of sulfate and nitrate anions, originated from mixed acids catalysts, (rendered from military ammunition plants))

  1. With reference to comment 11, since the Ag is deposited on the surface, the word "doped" must be changed throughout the manuscript to "loaded on the surface."

Ans: The sentence has been rewritten as reviewer instructed. (Page3, As loading amounts of Ag increased, more clumps of particulates were observed.)

  1. Regarding comment 12, the main purpose of the XPS analysis is to show the nature of Ag after preparation, not after the reaction, because it could be that the authors prepared Ag oxide instead of Ag metal.

Ans: (a) XPS analyses would provide evidence on the NiO semiconductors which could be activated by ultrasonic irradiation and generate sono-induced electron-hole pairs. Therefore, some Ni2+ was transformed into Ni3+, due to migration of sono-induced electrons. (b) The Ag metal particles on the surface of NiO have been verified by means of XRD analyses. (shown in Fig. 2)  

  1. Regarding comment 14, the authors should show the XRD analysis in the manuscript and include it in the discussion. Also, they should show in the graph the planes of Ag nanoparticles, which will confirm the formation of Ag nanoparticles. The diffraction peaks of Ag confirm the formation of the Ag nanoparticles, which can be used instead of the XPS analysis. Please refer to the following article and cite it in the text: "https://doi.org/10.3390/pharmaceutics14051104".

Ans: This paragraph has been rewritten as reviewer instructed. (Page 3, Figure 2 illustrates X-ray diffraction patterns of Ag(1-4wt%)/NiO semiconductors. The four particular diffraction peaks at 2q values of 37.5°, 43.8°, 63.9°, and 77.4°, corresponding to the planes of (111), (200), (220), and (311) respectively, possess characteristic peaks of face-centered cubic structure of metallic silver [58, 59]. ) The article mentioned has been cited in the text.

  1. Regarding comment 17, the authors should mention in the manuscript that this reaction intermediate is expected based on previous studies..

Ans: The sentence has been rewritten as reviewer instructed. (Page11, Obviously, hydroquinone was thought to be reactive intermediates of phenol, which would be successively transformed into p-benzoquinone via hydrogen abstraction, based on our previous studies [35, 36].)